# The Oxygen–Ozone Adjunct Medical Treatment According to the Protocols from the Italian Scientific Society of Oxygen–Ozone Therapy: How Ozone Applications in the Blood Can Influence Clinical Therapy Success via the Modulation of Cell Biology and Immunity

**DOI:** 10.3390/biology12121512

**Published:** 2023-12-11

**Authors:** Salvatore Chirumbolo, Luigi Valdenassi, Umberto Tirelli, Giovanni Ricevuti, Sergio Pandolfi, Francesco Vaiano, Antonio Galoforo, Fortunato Loprete, Vincenzo Simonetti, Marianna Chierchia, Debora Bellardi, Tommaso Richelmi, Marianno Franzini

**Affiliations:** 1Department of Engineering for Innovation Medicine, University of Verona, 37134 Verona, Italy; 2Italian Scientific Society of Oxygen–Ozone Therapy (SIOOT), High Master School of Oxygen-Ozone Therapy, University of Pavia, 27100 Pavia, Italy; luigi.valdenassi@unipv.it (L.V.); sergiopandolfis2@gmail.com (S.P.); vaiano.francesco@gmail.com (F.V.); antonio@galoforo.it (A.G.); dottf.loprete@gmail.com (F.L.); simonetti.vinc@gmail.com (V.S.); mariannachierchia@virgilio.it (M.C.); or info@ossigenoozono.it (T.R.); marianno.franzini@gmail.com (M.F.); 3Tirelli Medical Group, 33170 Pordenone, Italy; utirelli@tirellimedical.it; 4Department of Drug Science, University of Pavia, 27100 Pavia, Italy; giovanni.ricevuti@unipv.it; 5Magenta Clinical Service, 16125 Genoa, Italy; debora.bellardi@gmail.com

**Keywords:** ozone, ozone therapy, SIOOT, Nrf2, clinics, hormesis

## Abstract

**Simple Summary:**

In this article, we show the effect of ozonated blood, i.e., ozone used via major autohemotherapy, on fighting antibiotic resistance in post-surgical infected wounds and via minor autohemotherapy (ozonated blood) for the adjunct treatment of knee osteoarthrosis. Ozone exhibited the ability to reduce bacterial inflammation and bacterial sepsis, pain, discomfort and musculoskeletal disability in the studies presented herein.

**Abstract:**

Background. Ozone is an allotrope of oxygen whose use in medicine has rapidly grown in recent years. Ozonated blood allows for the use of ozone in a safe modality, as plasma and blood cells are endowed with an antioxidant system able to quench ozone’s pro-oxidant property and to elicit the Nrf2/Kwap1/ARE pathway. Methods. We present two clinical studies, a case-series (six patients) observational study adopting ozone as a major autohemotherapy and topical ozone to address infected post-surgical wounds with multi-drug resistant bacteria and an observational study (250 patients) using ozonated blood for treating knee osteoarthritis. Results. Ozonated blood via major autohemotherapy reduced the extent of infections in wounds, reduced the inflammatory biomarkers by more than 75% and improved patients’ QoL, whereas ozonated blood via minor autohemotherapy improved significantly (*p* < 0.001) WOMAC and Lequesne’s parameters in knee osteoarthritis. Conclusions. The models described, i.e., ozone autohemotherapy in wound antimicrobial treatment and ozonated blood in knee osteoarthrosis, following our protocols, share the outstanding ability of ozone to modulate the innate immune response and address bacterial clearance as well as inflammation and pain.

## 1. Introduction

Ozone (O_3_) is an allotrope of oxygen, which is usually known as a pollutant and a component of the terrestrial atmosphere [1,2,3], although the formation of ozone has also been reported in biological systems [4,5,6,7,8]. 

From a chemical point of view, ozone is a triatomic gas molecule composed of three atoms of oxygen linked with covalent bonds of about 1.278 Å ± 0.003 Å length and an average angle of 116° 49′ ± 30′ due to a resonance mechanism by oxygen π electrons [9]. Like oxygen, O_3_ behaves as a powerful pro-oxidant molecule, having yet a higher oxidizing ability than O_2_ [10]. 

However, O_3_ has a lower chemical stability than O_2_, and moreover, it is rapidly quenched by the antioxidant and scavenging systems present in the biological fluids and cells once it has entered an organism, such as during the process known as oxygen–ozone medical treatment [11]. In stable and normal conditions, the half-life of indoor gaseous O_3_ is between 7 and 10 min [12], whereas in water, O_3_ solubility depends on the temperature and the gas concentration; according to Henry’s law and in distilled water at room temperature, O_3_ half-life is about 28 min, whereas it rises to 53 min with 1 mM citric acid [13]. 

When introduced in the peripheral blood, during a process known as oxygen–ozone major autohemotherapy (O_2_-O_3_-MAHT), the gaseous mixture O_2_/O_3_, with ozone at 40 μg/mL (0.84 μmol/mL), is quenched within only 5 min by the antioxidant molecules, scavenging enzymatic systems and cell biochemical components present in blood, inasmuch as 78% O_3_ reacts with L-ascorbate forming dehydro-ascorbate and 20% of O_3_ reacts with uric acid forming allantoin, whereas only 2% of O_3_ targets lipids forming lipo-peroxides (LPOs), such as the alkenals 4-hydroxynonenal (4-HNE) and 4-hydroxyhexenal (4-HHE) [14]. 

Due to its strong oxidizing properties, which lead O_3_ to the formation of oxidized molecules (ozonides) from target biomolecules such as lipids (forming LPOs) and proteins (where ozone oxidates -SH groups or thiols), O_3_ is particularly dangerous if inhaled, as the lung is enriched with lipidic surfactants and their disruption, even by low doses of O_3_, completely compromises the pulmonary function [15,16,17]. In particular, ozone, even at the dose of about 2 ppm (2 μg/mL), damages the folding structure of the lung surfactant protein B (SP-B), leading to a defect in the ability to interact at the air–water interface via negatively charged glycerol phospholipids and therefore failing in sustaining the complex changes in alveolar volumes and shapes during the breathing cycle [18]. 

The use of ozone in the medical adjunct treatment is therefore restricted to few administration routes, i.e., parenteral injection, intramuscular or paravertebral injection, autologous blood infusion, rectal insufflation and topical treatment, despite some odd, controversial attempts via inhaled O_3_ was recently reported [19]. Furthermore, the doses of ozone used in the adjunct medical treatment should be within the buffer ability of the total antioxidant system (TAS) in the plasma, in the range 20–80 μg/mL O_3_ and below 160 μg/mL [14,20]. The various O_3_ administration routes and different O_3_ dosages used for the adjunct medical treatments have been debated and addressed in specialty sessions of experts joining a scientific society board, which deals with the better straightforward application of O_3_ in many medical fields [21]. 

A search on the medical scientific database Pubmed allows researchers and clinicians to retrieve 4504 items dealing with “ozone therapy” (on 19 October 2023), of which there are only 334 clinical trials, 237 randomized controlled trials (RCTs), 29 observational studies, 80 systematic reviews and 43 meta-analyses; the most are debating reports, extensive reviews and commentaries and in vitro or in vivo studies with cells and laboratory animals. 

The interest toward ozone as an adjunct treatment in medicine has increased from 120 scientific papers/year in 2001 to 309 in 2021, whereas the general interest worldwide for ozone in the medical therapy, as reported by Google Trends, has increased and joined more and more countries from 2004 to date (Figure 1).

It is clear that the introduction of ozone in medicine is envisaged as a formidable opportunity for clinicians, and in this overview, we try to provide insights into this widespread opinion and shed light on the role of ozone in the medical field.

Ozone in the blood (ozonated blood) is the best way through which to use ozone as a successful adjunct treatment, because it is rapidly quenched by TAS, leaving only bioactive intermediates such as 4-HNE and malonyl dialdheyde (MDA) and triggering the Nrf2 signaling system. In this report, we will show how ozonated blood, either via major autohemotherapy or minor ozonated blood infusion, is able to address challenging medical painful concerns such as infected post-surgical wounds and musculoskeletal disorders, for example knee osteoarthritis. 

## 2. Ozone in the Treatment of Infected Wounds and in Antibiotic Resistance

### Use of Topical Ozone and O_2_-O_3_ Major Autohemotherapy (O_2_-O_3_-MAHT)

Ozone can be easily applied to infected and ulcerous wounds as an antimicrobial and anti-inflammatory topical formulation [22]. This ability is due, respectively, to the antimicrobial effect of the pro-oxidant potential of ozone and to the anti-inflammatory activity held by hormetic ozone at a systemic level, and this encouraged the Italian Scientific Society of Oxygen–Ozone Therapy (SIOOT) in challenging these properties to address the huge concern of antibiotic resistance [23,24]. 

The use of ozone in eradicating multi-resistant bacteria (MRB), or multidrug-resistant (MDR) bacteria, particularly in nosocomial contexts, has been exclusively limited so far to using this oxygen allotrope as a disinfectant for indoor environments and wastewater [25,26,27,28]. Yet, low doses of ozone, along with oxygen in a balanced mixture of oxygen/ozone, are widely used as a successful therapeutic approach to hinder microbial infections and reduce or block the consequent immunity-driven inflammation in several forms of inflammatory disorders caused by bacterial infection [29,30,31,32,33,34]. Even the way by which ozone should kill bacteria inside organisms has been assessed by demonstrating the ability of innate immune cells to produce ozone in an antibody-dependent mechanism and by amino acids assistance [7,35]. In these cases, neutrophil’s exposition to the radical singlet molecular oxygen (^1^O_2_•) allows antibodies to oxidize water, then produce H_2_O_2_ via the intermediate H_2_O_3_ (trioxidane), a soluble and highly active form of ozone [36]. 

Despite ozone not being commonly used by innate immune cells to remove bacteria via phagocytosis, as other reactive species such as oxygen radicals and hypochlorite ions are used, ozone is widely considered as a powerful disinfectant and its application has been used also in clinics [37,38].

The SIOOT/Multiossigen protocol to address an infected wound, even if infected by MRB or MDR bacteria such as methicillin-resistant *Staphylococcus aureus* (MRSA), consists of a bimodal ozone treatment using topical ozone via ozonated water and ozonated oil and O_2_-O_3_-MAHT sessions. The strategy is based on the Janus hallmark held by ozone [39], inasmuch as ozone can directly kill bacteria invading a post-surgical or traumatic wound, even ulcerous wounds, or put to an end via hormesis to the systemic antimicrobial struggle held by the innate immunity. 

## 3. Ozone in the Treatment of Painful Inflammation, Disabilities and Fatigue—The Knee Osteoarthritis Model

### Use of Ozonated Blood via O_2_-O_3_ Minor Autohemotherapy (O_2_-O_3_-mAHT)

The best performance exhibited by the adjunct medical treatment with ozone, regards, as reported also in the evidence above, its ability to reduce and often removing pain, in addition to motor disability, discomfort and fatigue, commonly associated with chronic inflammatory and/or immune pathologies [40,41,42,43,44,45,46]. 

The ability of ozone to reduce pain might be attributed to its ability to reduce inflammation via the Nrf2/Keap1/ARE pathway and the inhibition of NLRP3 by 4-HNE (see above). A SIOOT protocol, using O_2_-O_3_-mAHT, to reduce pain and motor disability in knee osteoarthritis (OA) has been recently addressed. 

Gonarthrosis is a disused terminology, now known as knee osteoarthritis (knee OA), indicating a chronic disease developing at the joint level and with degenerative lesions affecting the articular cartilage which progressively cause pain, difficulty in movement and, in more severe cases, deformation of the joint itself [47,48]. Primary arthrosis affects all areas of the knee (panarthrosis) and is often associated with other locations of the pathology such as the hands, hips or spine. Secondary arthrosis, on the other hand, promotes a specific part of the joint [49,50,51]. 

Moreover, an alternative way to further classify knee osteoarthritis is also based on the location of cartilage wear (internal, external or patellar–femoral), and when the wear is limited to one side only, it is referred to as “single compartment” knee OA. Often, this is also the way in which the pathology begins, which will only later involve the other anatomic regions of the knee. In recent years, therapy of knee OA has included many infiltration methods in the knee synovial joint, using platelet-rich plasma (PRP), autologous serum or other autologous blood-derived practices [52,53,54,55,56,57]. 

Fundamentally, blood component infiltrations rely on the presence of various biomodulating factors, such as growth factors [52] notoriously present in PRP, e.g., platelet-derived growth factor (PDGF), epithelial growth factor (EGF) or vascular endothelial growth factor (VEGF) [58,59], in addition to immunomodulatory cytokines [52]. Humoral, cell-free components from peripheral whole blood rely on the acknowledged therapeutic activity of growth factors, cytokines, chemokines and other soluble immune mediators, whereas the cellular fraction is much less considered as a whole. 

Yet, recently, chondrogenic precursors and peripheral blood-derived stem cells were put in the spotlight, promoting peripheral blood as a primary source of therapy via infiltrations in the knee OA [60,61,62]. Infiltrating autologous peripheral blood, rather than a simple autologous conditioned serum or a PRP [63], may improve the ability to involve the activity of chondrocyte mesenchymal precursors from whole peripheral blood. Finally, some authors showed that medical ozone therapy may regenerate damaged articular cartilage in osteoarthritis [64]. 

The use of an oxygen–ozone mixture via autologous peripheral blood infiltration (ozonated blood via a kind of minor autohemotherapy) should replace the direct intra-articular infiltration of ozone, as major bioactive components of ozone are actually represented by blood-derived electrophiles such as 4-hydroxynonenal (4-HNE) or lipo-peroxides from polyunsaturated fatty acids (LOPs) [65,66,67,68]. Furthermore, ozone may even positively affect cellular regeneration and stem cell biology [69,70], therefore suggesting its promoting, rejuvenating and reconstructing role when present in the whole blood and infiltrated in the knee. 

Starting from this evidence, we addressed knee OA in a cohort of patients using ozone in the whole blood and infiltrating autologous ozonized blood (O_3_-WB) in the knee using the SIOOT protocol from an original protocol proposal by Prof. Luigi Valdenassi, who held the research study.

## 4. Materials and Methods

### 4.1. Patients’ Recruitment in the Ozone-MDR Study

Considering that in the present study, we expected an anticipated incidence in reducing pain and inflammation by antimicrobial therapy of 90% (group before treatment) and an anticipated incidence in reducing pain and inflammation by ozone of the same entity (90%) (group after treatment), with a Type I/II alpha error = 0.05 and a power of 80%, a number of 10 samples is the minimum to reach a sound statistic in a binomial option “ill-not ill”. Cohen’s d test = 0.45929, indicating a medium effect size, which is to be considered a possible initial limitation of this pilot study. Four samples were excluded because they did not complete the scheduled therapy.

Outpatients resorting to the Comunian Clinics (Gorle, BG, Germany) were randomly selected from within a wider population having the following major eligible criteria at the time of the study, independently from sex, race, BMI and age distribution: (a) presence of recurrent bacterial-driven inflammatory episodes failing to be treated with available antimicrobials; (b) ascertained antibiotic resistance with present bacterial-associated inflammation; (c) never being previously subjected to oxygen–ozone therapy; (d) without chronic pathologies such as cancer or neurodegenerative and behavioral disorders; (e) washed out from taking pain relief drugs for at least 72 h. 

Patients signed an informed consent form according to the Declaration of Helsinki and the Ethical Committee of SIOOT.

### 4.2. Protocols of Ozone Therapy

The protocols entailed a minimum of five runs of two weekly O_2_-O_3_-MAHTs (50 μg/mL in 200 mL of autologous blood), supported with one microinjection/week of 5 μg/mL O_3_ in situ and/or by rectal ozone insufflation (100 mL of 40 μg/mL O_3_) and/or topical 5% ozone in a dermatologic formulation. The final amount of O_3_ into 200 mL of blood should consider Henry’s law; so, according to our calculations based on the Multiossigen s.r.l. (Bergamo, Italy) technology, which ensures the reliability of O_3_ quality production, actual O_3_ in 200 mL is ~1.586 mg at 37 °C, pH = 7.4. The application of rectal ozone was excluded for patients showing gut microbiome dysbiosis and rectal pathologies. Its use was recommended in cases of impairments in the mucosal immune response and to enhance the participation of gut microbiome in the immune surveillance [71,72]. 

The protocols followed the criteria of the International Scientific Society of Oxygen–Ozone Therapy (SIOOT), with the technological support of Multiossigen s.r.l. Gorle (BG). For the oxygen–ozone, any treatment via autologous blood venous infusion (O_2_-O_3_-MAHT) needs an ozone generator and medical-grade compressed oxygen, in addition to a CE-certified SANO3 blood collection sterile bag and a device for hemotransfusion, including a syringe. A volume of 200 mL of peripheral blood was withdrawn from any patient via venipuncture, collected in the bag and gently mixed with the prepared oxygen–ozone mixture (Multioxygen Medical 95 CPS™, from Multiossigen s.r.l., Gorle (Bergamo, Italy), containing 50 μg/mL ozone (O_3_) and within 2 min re-introduced it into the patient’s blood circulation via an intravenous cannula.

### 4.3. Evaluating Patients’ Outcomes: Inflammation and Antimicrobial Biomarkers

Each patient recruited in this study was investigated for the major plasma inflammatory biomarkers (CRP and ESR) and for blood cell counts via routinely clinical chemistry and hematological analyzers used in the various hospitals where patients were hospitalized and/or visited. Visual signs and imaging of tissue inflammation associated with traumas or other damages were also recorded. The most exemplificative visual cases are reported in the Results section and further discussed. Each patient was evaluated for the antimicrobial sensitivity/resistance by MIC assay (see also Table 1).

### 4.4. Evaluating Patients’ Outcomes: The McGill Quality of Life (QoL) Questionnaire

To evaluate patients’ QoL following our ozone therapy protocols, an updated form of the McGill QoL questionnaire was used [73,74,75] with patients before accessing the O_2_-O_3_-MAHT and at the end of the therapy cycle (within one week later). The questionnaire evaluated three parameters, from the minimum to the maximum, namely: the Pain Rating Index (PRI), including a scale from 0 to 78; the Pain Rating Index referring to Time (PRIT), with a scoring from 0 to 20; and the Present Pain Intensity (PPI), with a scoring scale of 0–5. Values were statistically analyzed with a Kruskal–Wallis test for *p* < 0.05. 

### 4.5. Statistics

Mean ± standard deviation (SD) and medians were calculated for raw data entering our research study and analyzed with a Wilcoxon’s signed rank test for non-parametric distributions, once the Kolmogorov–Smirnov (KS) test had assessed a non-normally distributed sampling. When necessary, data were plotted (Sigma plot v.14.0 software) accordingly. For score data, a Kruskal–Wallis test (*p* < 0.05) was performed. An SPSS v 26.0 software was used for this purpose. The relative bias in percent, to assess if the discussion of the data could be biased, was calculated according to the following formula:$$\mathrm{RE}=\frac{\sum_{i=1}^{n}(P_{i}-O_{i})}{\sum_{i=1}^{n}O_{i}}$$

This bias is expressed in relative terms (relative bias) and makes it possible to evaluate the size of the bias due to under-coverage with respect to the true unknown parameter to be estimated. 

### 4.6. Patients: Study on Knee OA

A number of 265 clinical outpatients (mean age 63.2 ± 8.3 SD years old, 71% females) asking for ozone therapy in our clinical health services were screened for enrollment in the study. Fifteen patients, for various reasons, did not reach the therapy completeness and therefore were excluded from data mining, elaboration and result reporting, while 250 entered and completed the study. Each patient signed an informed consent form and underwent any ethical criteria in compliance with the Helsinki Declaration (9 July 2018). The study was conducted by the research staff held by Prof Luigi Valdenassi, in Genoa and Gorle (BG), at SIOOT. 

The major eligible criteria were the presence of knee OA according to the Ahlback classification [76]. Exclusion criteria were patients with a knee OA Ahlback score ≥ 3, articular damages or traumas not related to knee OA, both knees suffering, varus or valgus deformity ≥ 10 degrees, flexion contracture higher than 15 degrees, ligament or meniscus impairments, up-taking pharmaceutical drugs or NSAIDs in the previous 30 days, knee infiltrations in the previous 12 months, surgical interventions in the previous 6 months, physiotherapy in the previous 30 days, and further chronic systemic immune or autoimmune disorders, cancer or active infections. Upon admission, patients were interviewed for their knee OA symptomatology description and gave their signed consent to join a QoL questionnaire once the therapy program was completed.

The study was performed on 250 patients, 173 female (62.4 ± 2.4 SD years) and 77 male subjects (64.1 ± 3.1 SD years), of which 189 (134 females) were classified as grade 1 Ahlback knee OA and 61 (35 females) were classified as grade II. 

### 4.7. Whole Blood Ozonation in Knee OA

In this study, we used the ozone-producing device Multiossigen^®^ Medical 95 (Multiossigen s.r.l. Gorle (BG), Italy), which is an outpatient bed unit for routine oxygen–ozone major autohemotherapy (O_2_-O_3_-MAHT) that is able to ozonize biological liquid at disposal. The machinery allows physicians and operators to customize the O_2_-O_3_ gas mixture according to the clinical request, i.e., the clinical therapy protocols, and to withdraw the requested mixture of O_2_-O_3_ at disposal. The same device can ozonize autologous blood for minor autohemotherapy.

The equipment joined with the device was for medical and/or research use only. The O_2_-O_3_ mixture generator was tuned and regulated by an adaptive microprocessor, which ensures the precision of the ozone delivery, once the O_3_-O_2_ mixture amount was selected by the operator, via fluorometry detectors. The machinery can to customize the therapeutic treatment by selecting the ozone concentration in a continuous range, namely, from 1 to 100 μg of O_3_. Moreover, the ozone concentration can also be modulated by further varying the oxygen percentage of the mixture. Any treatment can be performed by only expert and highly trained medical personnel (II level graduated; Master’s) and can be performed in all conditions of use, even for topical treatments.

An autologous withdrawal of 2.5 mL venipuncture Na-citrate anticoagulated peripheral blood with a 5.0 mL syringe was gently mixed with 2.5 mL of a O_2_-O_3_ mixture (20 μg/mL O_3_ final *v*/*v*) and injected in the knee, according to the description provided in the next paragraph. 

### 4.8. Patient’s Knee Infiltrations

The infiltration of the 5.0 mL autologous blood mixture (20 μg/mL O_3_) was usually performed (in compliance with our protocols) in various periarticular points, previously marked with a pencil, and by previously disinfecting the inoculation point. The operator used a frontal access for infiltrations, having the patient in a supine position and the knee bent at 90°. The reference points were represented by the patellar ligament, the lower edge of the femur condyle and the upper edge of tibia plane. The insertion of the 23 G needle was put slightly upward and parallel to the tibial plane, moderately inclined medially towards the intercondylar pit. Each patient underwent 1 infiltration a week for a total of 5 weeks. 

### 4.9. WOMAC Index and Lequesne Algofunctional Index

Patients were interviewed before the infiltration treatment, on beginning the research study, and following one week from the last infiltration via the WOMAC (Western Ontario and McMaster University) Osteoarthritis index, which is a PRO (patient-reported outcome) questionnaire that aims to identify variations in symptoms and activity restrictions not only in subjects with hip and knee osteoarthritis but also in subjects undergoing replacement surgery [76]. The WOMAC index evaluates patient’s pain (5 items, score 1–20), functional limitations (17 items, score 0–68) and stiffness (2 items score 0–8). Furthermore, patients were followed up with a Lequesne algofunctional index [77]. 

### 4.10. Statistics 

Data were expressed as mean ± standard deviation (SD) and inferred with a non-parametric Wilcoxon rank test at *p* < 0.05. Graphs were plotted using a Sigma plot v 14.0 and elaborated with SPSS v.24. A Kruskall–Wallis (KW) test was applied also for a comparison of score statistics. The sample size, calculated within the margin of error ≤7.0% and a default population proportion of 50%, was estimated as 196 patients, for a confidence interval of 95%; therefore, 250 patients, from January 2019 to November 2021, give a Cohen d = 0.0134 and an effect size of 0.067, meaning that no statistical confounders regarding patients’ distribution were present. 

## 5. Results

### 5.1. Major Ozone Autohemotherapy along with Topical Ozone Reduces Infection and Inflammation in Post-Surgical Wounds

Table 1 summarizes the results of six cases of patients showing post-surgical wounds infected with MRSA before and after the SIOOT/Multiossigen protocol of ozone therapy. Before resorting to the specific ozone therapy approach recommended for each single individual, patients showed a previous clinical history characterized by a marked increase in the major inflammatory biomarkers, such as C-reactive protein (CRP) (41.14 ± 3.25 SD mg/dL), erythro-sedimentation rate (ESR) (56.11 ± 12.14 SD mm/h) and PCT (0.69 ± 0.32 SD μg/L), along with alterations in the white blood cells number and sometimes in the plasma level of fibrinogen. 

Due to the heterogeneity in the bioanalytical panels recommended by the different clinics and hospitals, where the patients recruited in this study had received previous care, we were able to select only the major and commonest inflammatory markers to assess that any patient recommended to ozone therapy was still in active inflammation. A calculation of the relative bias was performed.

Following O_2_-O_3_-MAHT, any inflammation marker disappeared, resulting in parameters returning to their healthy normal ranges quite immediately, at the second ozone treatment, as the biomarkers rapidly normalized their plasma levels. CRP dropped down from 41.14 ± 3.25 SD mg/dL to 0.47 ± 0.02 SD mg/dL (−98.86%, *p* = 0.0027), ESR from 56.11 ± 12.14 SD mm/h to 11.30 ± 1.25 SD mm/h (−79.86%, *p* = 0.0012) and PCT to ≤ 0.05 μg/L (*p* < 0.001). The reduction in the inflammation impact associated with post-traumatic sepsis or bacterial colonization via a specific antimicrobial therapy failed in all of the patients treated before ozone therapy as a result of continuous relapses in inflammation events and a modest reduction in the major biomarkers (CRP = −16.16%, *p* = 0.00512; ESR = −10.54%, *p* = 0.0053) due to MRB (Table 1). The RE value for major quantitative data was −7.3% (CRP) and −10% (ESR), which is within the tolerability index for data distribution acceptance [78]. For PCT values, they were rather inhomogeneous and were not further elaborated.

Pain and discomfort, assessed by a modified McGill questionnaire, were almost completely reduced following the ozone therapy. The Pain Rate Index (PRI) dropped down from 69.9 ± 6.66 SD to 23.6 ± 7.99 SD (−66.23%, *p* = 0.00016), the Pain Rate Index referrning to Time (PRIT) decreased from 18.6 ± 1.43 SD to 7.30 ± 2.11 SD (−60.75%, *p* = 0.00021) and the Present Pain Intensity (PPI) was reduced from 4.60 ± 0.70 SD to 1.00 ± 0.66 SD (−78.26%, *p* = 0.00018) (Figure 2).

### 5.2. The Effect of the Ozonated Blood in the Minor Autohemotherapy on Knee OA

Figure 3 shows the effect of five runs of ozonized blood in knee OA-affected patients (Figure 3 panel A1: female; and Figure 3 panel A2: male) showing a type I Ahlback knee OA. In female patients, pain was reduced by five score units (from 10.45 ± 2.89 SD to 5.12 ± 1.49 SD, *p <* 0.0001), functional deficit by thirteen score units (from 22.60 ± 2.34 SD to 12.30 ± 2.39 SD, *p <* 0.0001) and stiffness by three score units (from 6.30 ± 2.18 SD to 2.85 ± 1.70 SD, *p <* 0.0001), i.e., ozonated blood roughly halved pain, functional deficit and stiffness (51.01%, 45.58% and 54.81%, respectively). 

Male patients with type I Ahlback knee OA showed comparable results: pain was reduced by three score units (from 10.42 ± 1.25 SD to 5.00 ± 1.56 SD, =−52.03%, *p* < 0.0001), functional deficit by nine score units (from 19.06 ± 1.89 SD to 10.00 ± 1.60 SD, =−47.53%, *p* < 0.0001) and stiffness by two score units (from 4.48 ± 1.87 SD to 2.06 ± 0.75 SD, = =−54.05%, *p <* 0.001). Female patients showed a better response to ozonized blood therapy (delta score = +2.3 score units, KW test *p* = 0.00311) than males. 

Figure 3 panel B shows the effect of five runs of ozonized blood in knee OA-affected patients (Figure 3, panel B1: female and Figure 3, panel B2: male) showing a type II Ahlback knee OA. In female patients, pain was reduced by nine score units (from 21.51 ± 3.42 SD to 12.51 ± 1.52 SD, *p <* 0.0001), functional deficit by nineteen score units (from 38.64 ± 6.65 SD to 20.42 ± 2.93 SD, *p <* 0.0001) and stiffness by five score units (from 9.33 ± 2.66 SD to 4.24 ± 1.68 SD, *p <* 0.0001), i.e., even for type II Ahlback knee OA, ozonated blood roughly halved pain, functional deficit and stiffness (41.83%, 47.13% and 54.54%, respectively). Male patients with type II Ahlback knee OA showed, as with type I, comparable results. Pain was reduced by eleven score units (from 16.81 ± 3.39 SD to 6.51 ± 2.17 SD, = −61.26%, *p* < 0.0001), functional deficit by sixteen score units (from 35.24 ± 6.09 SD to 19.42 ± 3.07 SD, = −44.88%, *p* < 0.0001) and stiffness by four score units (from 7.81 ± 3.12 SD to 4.51 ± 1.25 SD, = = −42.25%, *p* < 0.0001). 

Apparently, males with type II Ahlback knee OA responded better in the WOMAC pain score than females, but globally, type II Ahlback knee OA females confirmed, even if in a slighter way, their best response to WOMAC ranking than males. But this difference was not significant (delta score = +1.3, KW test *p* = 0.45977). 

Figure 3 panel C shows the Lequesne’s index for knee OA severity, as affected by ozonated blood infiltrated in the knee. In type I Ahlback knee OA, the Lequesne’s index in females dropped down from 9.42 ± 2.91 SD to 4.21 ± 2.45 SD (*p <* 0.0001, Figure 3, panel C1), i.e., by 55.31%, whereas in males, the ozonized blood reduced the Lequesne’s index from 9.48 ± 2.45 SD to 4.33 ± 2.38 SD (*p <* 0.0001, 48.82%, Figure 3, panel C2). In type II Ahlback knee OA, ozone reduced the Lequesne’s index in female patients from 17.24 ± 5.00 to 8.36 ± 2.13 SD (*p <* 0.0001, 51.51%, Figure 3, panel C3), whereas in males, ozone infiltrated as ozonized blood in the knee reduced the Lequesne’s index from 14.82 ± 2.69 SD to 5.28 ± 2.62 SD (*p <* 0.0001, 64.38%, Figure 3, panel C4). 

In type II Ahlback knee OA male patients, ozone appears to be much more effective than in females and in type I Ahlback knee OA males. A total of 154 patients (85 females) were interviewed for pain, stiffness and quality of life and did not report any discomfort, pain or fatigue after a minimum period of 8 months. 

## 6. Discussion

### 6.1. Ozone in Major Autohemotherapy against MDR Bacteria

Our interpretation of the reported data relies on two modalities with which ozone exerts its antimicrobial action. Ozonated formulations can directly kill bacteria in infected wounds [79,80,81,82].

In this circumstance, it is paramount to clarify that topical ozone acts via its chemical ability to cleanse wounds by killing bacteria due to lysis induced by peroxide derivatives. Furthermore, ozone in low doses via peripheral blood exerts a major role on the host’s immune response. For example, ozone used in many oxygen–ozone therapy approaches, such as major autohemotherapy (O_2_-O_3_-MAHT), targets inflammatory mechanisms by acting on more complex intracellular signaling pathways, modulating the interplay between Nrf2/NF-κB and the mitochondria-associated inflammasome NLRP3, occurring for viral diseases and other inflammatory pathologies [83,84].

Ozone activates the Nrf2/Keap1/ARE signaling pathway, which induces the activation of heme oxygenase-1 (HO-1) and the production of carbon monoxide (CO) from degraded heme groups. Carbon monoxide inhibits the activation of NF-κB, and therefore of the inflammatory signal, and promotes the macrophage skewing of M1 to M2 [85,86]. Systemic ozone, provided via O_2_-O_3_-MAHT, enhances the ability of innate immune cells, as the activation of Nrf2 contributes to shifting the presence of M1 toward skewing to the M2-phenotype, which are “trained” innate cells to improve bacterial and cell debris phagocytosis [87,88,89]. 

Moreover, the activation of Nrf2 by systemic ozone increases the expression of the macrophage receptor with a collagenous structure (MARCO), which, if overexpressed, enhances the ability of M2-macrophages to phagocytose bacteria and promotes the interplay between innate and acquired immunity, so as to improve the complete clearance of infectious bacteria [90,91]. In addition, ozone elicits the production of 4-hydroxynonenal (4-HNE) from ω6-PUFAs, an alkenal that inhibits the activation of the inflammasome NLRP3 and the subsequent maturation of the pro-inflammatory cytokines IL-1ß and IL-18 [92]. We reported the ability of ozone, often used in a multi-step therapy approach, to overcome the exacerbating process of inflammation in patients with chromicized wounds (usually post-surgical wounds) due to colonization from bacteria in an MDR circumstance, i.e., MRSA. All patients considered exhibited the presence of at least three antibiotic-resistant strains in their antibiograms. 

Ozone showed the ability to promote almost complete healing of these patients upon whom the recommended antibiotic therapy did not work optimally. Furthermore, the major inflammatory markers and QoL questionnaires reported a scenario of recovered health in those patients with antibiotic resistance when undergoing ozone therapy. The SIOOT protocol to eradicate bacteria-caused inflammation in post-surgical patients with antibiotic resistance patterns is grounded on two/three fundamental mechanisms (Figure 4).

When the microbial involvement of the surgical wound of any other traumatic wound is particularly widespread, chemical ozone, either in the form of ozonated injected solutions or topical formulas, rapidly quenches bacterial viability and growth, then facilitating the host’s immune system to overcome the antimicrobial resistance [93]. Medical ozone, used via the major oxygen–ozone autohemotherapy (O_2_-O_3_-MAHT), modulates the patient’s immunity, leading to the complete eradication of the septic mechanism via the immune response. 

The modulating activity of O_2_-O_3_-MAHT may be implemented by adjusting the contribution of the gut microbiome/gut immunity interplay and the mucosal immunity (IgA) via the use of rectal ozone [72]. The first approach (chemical) is to be considered as an ancillary approach of the leading way of action, which is held by O_2_-O_3_-MAHT. Evidence about the ability of topical ozone solutions, ozonated oils and other topical formulas to remove antibiotic-resistant bacteria from the skin, mucosae or eyes has been widely reported in the literature to date [94,95,96,97]. In a recent paper, some authors investigated the action of an ophthalmic solution containing ozonated oil (10.5%, i.e., 5 μg/mL) against 60 bacterial strains with an MDR profile for various patients (resistance was confirmed for at least three different antibiotics, including fluoroquinolones) and showed that at 6–8 h following the treatment, at least 94% of all strains, i.e., *Pseudomonas aeruginosa* MDR, methicillin-resistant *Staphylococcus aureus* and methicillin-resistant *Staphylococcus epidermis*, reported a marked inhibitory zone and that more than 10% of those strains were also inhibited after 24 h [97]. 

However, bacterial growth may not be completely dampened by only topical or injected ozonated solutions in the neighboring skin of the wound [97]. This should assess the recommendation to include an O_2_-O_3_-MAHT as the leading therapy approach to overcome the possibility of a bacterial relapse [97]. Ozone acts in the blood in a completely different way with respect to the direct contact of the ozonated solutions and formulas [14,98,99]. As the contact of low doses of ozone in the peripheral blood activates the formation of lipo-peroxide intermediates and electrophiles, such as 4-hydroxynonenal (4-HNE) [10,92,100], this intermediate may reduce the impact of the inflammasome (NLRP3) role in the septic inflammation and inhibit innate cell pyroptosis, thus acting as an anti-inflammatory compound and immunomodulating the [10,92] inflammation process [71]. 

The concurrent action of the topical ozone, working in a quite direct way, and of blood-derived ozonized products, such as 4-HNE, which acts as a modulatory agent, should warrant for the complete effectiveness of ozone in the clearance of MDR bacteria in humans. Moreover, the ability of ozone via autohemotherapy to elicit a “mito-hormetic” mechanism [14,21,23,24,99] may explain how ozone has the ability to restrain inflammation by inducing a macrophage-tolerant state in an LPS-induced immune exacerbation [71].

### 6.2. Minor Autohemotherapy with Ozonated Blood Reduces Discomfort and Disability Markers in Knee OA

Recent reports assessed the use of ozone in knee OA [101,102,103]. Regarding the excellent clinical trial by Lopes de Jesus et al. [102], we did not use directly a gaseous oxygen/ozone mixture in the infiltrated knee but a stabilized medium made of ozonated blood instead, whilst maintaining the ozone dosage at 20 μg/mL. This approach should promote the formation of ozonized mediators, such as 4-HNE or LOPs from fatty acids, eliciting a hormetic or mito-hormetic mechanism in the knee tissue compartments [14,21,66]. 

Lopes de Jesus and coworkers obtained encouraging results by using ozone directly in the knee but used 1.5 fold more infiltrations (eight instead of 5five and obtained a reduction of about 50% in WOMAC or Lequesne’s scoring, the same as our results. In our hypothesis, the strong oxidative potential of ozone directly in the knee might reduce the hypoxic microenvironment, which causes severity in the knee osteoarthritis, thus relieving the concerning symptoms [102,104,105]. 

The hormetic mechanism triggered by ozone-caused lipid mediators and electrophiles allows the modulated reduction in the hypoxic microenvironment [106], with the paramount difference that reactive oxygen species (ROS) do not increase thanks to an improvement in the mitochondria activity, whereas in the pro-oxidant activity of gaseous ozone directly, mitochondria burst a huge increase in the oxidative activity [106]. The hormetic modulation of the Nrf2/Keap1/ARE pathway by ROS as signaling molecules can occur via modulated involvements of mitochondria biogenesis (“mito-hormesis”) [107], whereas the pro-oxidant activity of ozone directly infiltrated elicits the activation of the antioxidant enzyme system but, at the same time, triggers a pro-inflammatory response of the cellular systems, for example NF-κB and inflammasome, therefore reducing the length of positive outcomes achieved with using ozone infiltrations and/or asking for more ozone to be used.

Our results, which enlarge previous data with less patients, suggest that ozone within blood should work in a better, much more suitable, affordable and physiological way than gaseous oxygen–ozone directly infiltrated in the knee.

Further insights are needed to elucidate the way in which ozone reduces dramatically the pain, stiffness and severity of knee OA. 

Autologous ozonated whole blood infiltrations in knee OA is able to reduce WOMAC indexes of pain, stiffness and functional disability, as well as Lequesne’s functional indexes, by at least 50% compared to the previous pathological condition. This evidence was attained with a reduced impact of infiltrating ozone from other current studies and represents encouraging evidence for the future of knee OA therapy. 

### 6.3. The SIOOT Protocol of Ozonated Blood 

The Italian Scientific Society of Oxygen–Ozone Therapy (SIOOT) arranged their ozone adjunct treatment protocols principally on the basis of the ozone interplay with the autologous peripheral blood, more than with other media which yet remain commendable as well. The main tenet of SIOOT methodology takes into account the fundamental action that ozone plays in the blood microenvironment, which consists of its ability in inducing PUFAs-derived alkenals (4-HNE and 4-HHE) and other LPOs, exerting a fine, modulatory activity on immunity and therefore on inflammation-related symptoms, such as pain and fatigue. In this manuscript, we collected two experimental examples of how autologous blood treated with ozone, either via O_2_-O_3_-MAHT or a minor autohemotherapy using ozonated blood, reduces inflammation-related events, such as infected wounds, chronic musculoskeletal inflammation and pain. In our opinion, an underlying shared mechanism can be highlighted. 

Ozone doses should not overcome the blood ability in buffering the pro-oxidant activity of this oxygen allotrope. Doses must be produced with sharpness and a background of high-quality technology, with reliable throughput to ensure the correct O_2_/O_3_ proportion is made in compliance with SIOOT protocols, which is crucial to properly use ozone in medicine [11,108,109]. 

While it may be understandable that ozone can be used to sanitize and cleanse infected wounds, its application is wider than expected. By eliciting the activation of the Nrf2/Keap1/ARE signaling pathway, i.e., acting as an antioxidant molecule, ozone exerts an anti-inflammatory effect. This is true for many ailments and pathologies of musculoskeletal etiology, for example low back pain and disc herniation [11,110,111,112,113], where ozone is used as a paravertebral/intramuscular injection, rather than intradiscal, which SIOOT recently criticized [114]. It is tempting to speculate that gaseous ozone (usually 30 μg/mL), once inside the skin tissue and muscles, is rapidly quenched by the antioxidant endowment of cells and induces the formation of LPOs and ROS as signaling molecules, triggering the activation of the Nrf2/HO-1/CO axis and probably inducing a microvasculature change due to the interplay between Nrf2/angiogenesis [115,116]. The anti-inflammatory activity of ozone and its mediators LPOs (mainly 4-HNE) promotes the skewing of M1 to M2 in the intervertebral disc during a disc herniation [117]. The M2 replacement enhances the renewal of nucleus pulposus, its ECM, cell survival and the remodeling of the external fibrosus annulus, as ozone can induce the synthesis of collagen I [118,119]. 

The ability of ozone to work as an anti-inflammation agent via hormetic mechanisms [14,21,98,100] is closely linked to the ability and expertise of clinicians to use ozone in a more effective way; for example, high doses should be avoided [120]. In this context, the numerous scientific societies, for example SIOOT in Italy, are always engaged and encouraged in tailoring the best medical approach to successfully use ozone in the adjunct medical treatment. 

The capability of ozone to modulate immunity, provided by physicians with a formidable tool, by addressing autoimmune disorders, such as multiple sclerosis [121,122], Sjögren syndrome [123], fibromyalgia [41] and ME/CFS [42], where ozone resulted in being particularly effective in reducing pain, discomfort, fatigue and other burdensome symptoms, has been shown. It is tempting to speculate that the immune-modulatory action held by ozone targets mitochondria, which are particularly involved in autoimmunity [124,125]. 

Once the gaseous mixture O_2_/O_3_, for example O_3_ = 45 μg/mL, enters the bloodstream via O_2_-O_3_-MAHT, a moderate hemolysis occurs, with release of heme byproducts (activating HO-1 and CO functions) and damaged erythrocytes, which are rapidly removed by O_3_-produced 4-HNE via eryptosis [126]. The gas O_3_ concentration into the plasma should be re-adjusted considering Henry’s law so that 45 μg/mL O_3_ (gas, r.t.) corresponds to about 7.165 μg/mL in water at 37 °C, a concentration which could yield 3.67 μM 4-HNE if only 2% of ozone is converted into 4-HNE [14,66,127]. A dose of 3.0 μM 4-HNE exerts an anti-inflammatory action [92]. 

Ozonate blood represents a straightforward possibility to apply the antioxidant and anti-inflammatory properties of ozone in medicine, yet ozone therapy should only be administered by trained and licensed practitioners in a controlled medical setting.

## 7. Conclusions

The use of ozone in autologous blood, which must be approached only by highly skilled experts and professionals, allows clinicians to use this oxygen allotrope as a sound therapeutical tool to address immune disorders (auto-immunity driven illnesses), musculoskeletal disorders with inflammation, infected post-surgical or traumatic wounds and many further inflammation pathologies, reducing the associated pain, fatigue, discomfort and disability, and exhibiting properties which encourage its consideration as an opportunity for medicine. 

## 8. Acronyms

Å: angstrom; 4-HHE: 4-hydroxyhesenal; 4-HNE: 4-hydroxynonenal; knee OA: knee osteoarthritis; Nrf2: nuclear factor erythroid 2–related factor 2; O_2_-O_3_-MAHT: major autohemotherapy with oxygen–ozone; O_2_-O_3_-mAHT: minor autohemotherapy with oxygen–ozone; SIOOT: Italian Scientific Society of Oxygen–Ozone Therapy; WOMAC: Western Ontario and McMaster Universities Arthritis Index.

## Figures and Tables

**Figure 1 biology-12-01512-f001:**
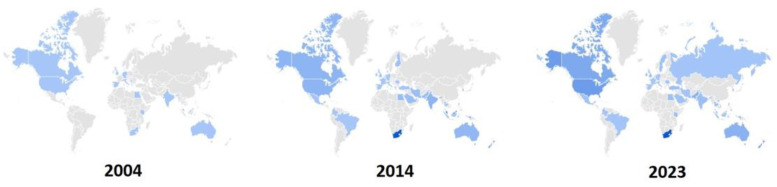
World maps representing the increasing number of countries (light blue) expressing interest in the ozone therapy according to the search in Google Trend of the term “ozone therapy” for the indicated years.

**Figure 2 biology-12-01512-f002:**
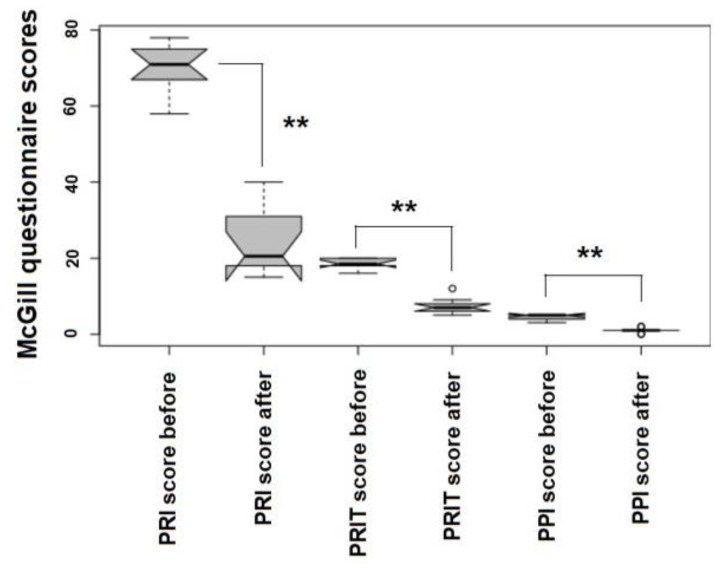
McGill questionnaire scoring values plotted as notched box plots. Statistics: ** *p <* 0.01 (Stata software and SPSS v 24.1).

**Figure 3 biology-12-01512-f003:**
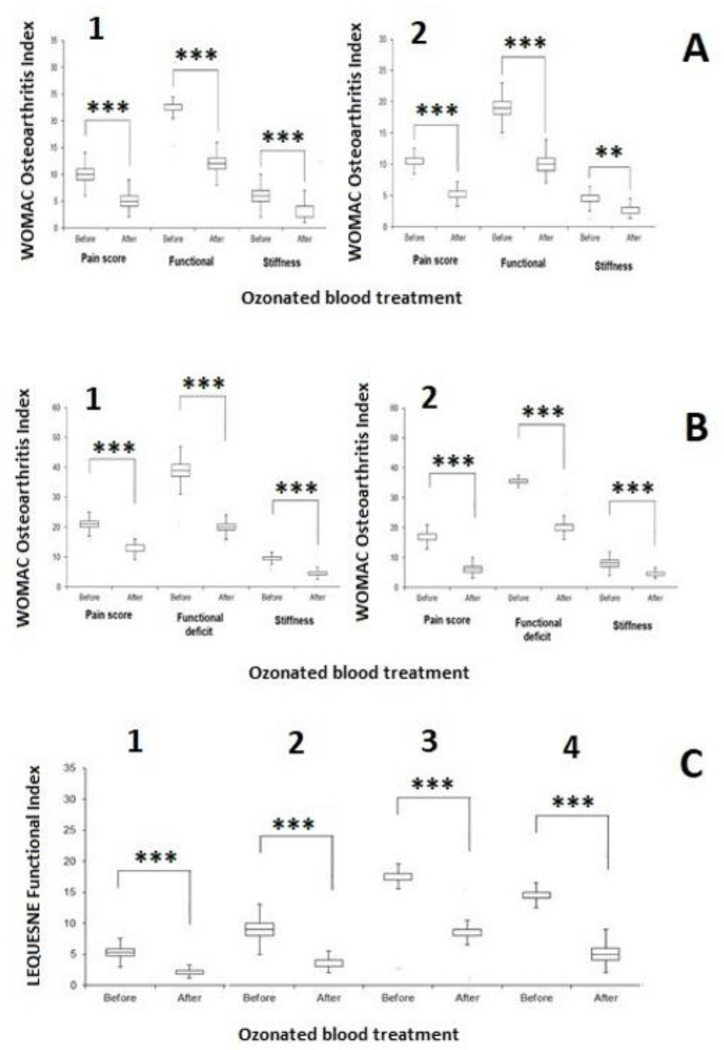
Box plot of the questionnaire scoring of patients with knee OA treated with ozonated blood. (**A1**) type I Ahlback knee OA female subjects: (**A2**): type I Ahlback knee OA male subjects; (**B1**): type II Ahlback knee OA female subjects; (**B2**): type II Ahlback knee OA, male subjects; (**C1**) Lequesne FI in type I Ahlback knee OA, females; (**C2**) Lequesne FI in type I Ahlback knee OA, males; (**C3**) Lequesne FI in type II Ahlback knee OA females and (**C4**): Lequesne FI in type II Ahlback knee OA, males. ** *p* < 0.001; *** *p <* 0.0001.

**Figure 4 biology-12-01512-f004:**
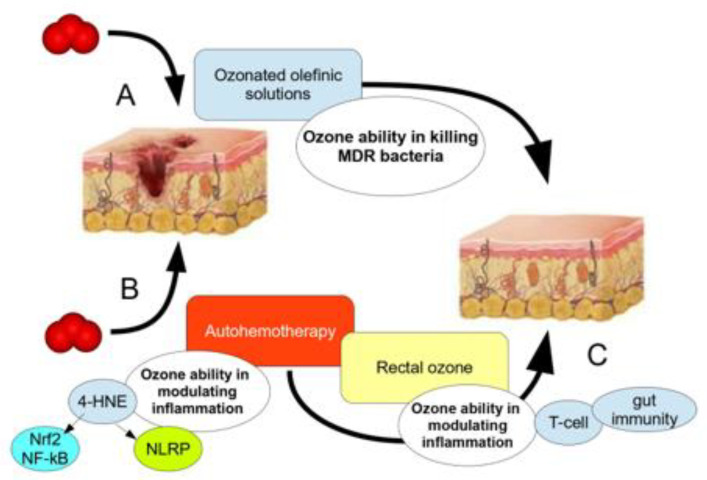
The ability of ozone to counteract the progression and exacerbation of infected wounds is based on three possible levels: (**A**) the direct chemical activity of ozonated formulations on the infected skin; (**B**) the immune regulation held by Nrf2 and 4-HNE, driving the M1/M2 skewing and promoting an anti-inflammatory mechanism, thus improving bacterial clearance; (**C**) (less investigated) the immune activity held by gut immunity.

**Table 1 biology-12-01512-t001:** Oxygen–ozone therapy (O_2_-O_3_-MAHT) outcomes in the recruited outpatients with inflammatory diseases.

Patient		Date of Birth	Clinical Condition in the Entrance	Microbiology Before After	Antibiotic Resistance	O_2_-O_3_-Maht	Outcome
1	JT	28 August 1964	14 February 2018. Pneumonia with chronic foci; left lung resistant to antimicrobial therapy; CRP = 13.50 mg/dL, ESR = 38 mm, WBC = 12.40 × 10^3^/μL	*S. aureus* 3.0 × 10^6^ CFU/mL 0.5 × 10^2^ CFU/mL	MRSA Failure to levofloxacin 500 mg + 250 mg 7 days	Five sessions of O_2_-O_3_-MAHT (1)	15 May 2019. Complete eradication of parenchymal foci.
2	GA	24 February 1969	4 March 2019. Bacterial infection of shoulder prosthesis.	*S. aureus* 4.7 × 10^7^ CFU/mL 3.2 × 10^1^ CFU/mL	MRSA	Five sessions of O_2_-O_3_-MAHT (1)	6 May 2019. Inflammation and infection disappeared.
3	AC	27 July 1975	8 April 2019. Osteomyelitis process following a tibial malleolar fracture with joint effusion; oedema in neighboring soft tissues; inflammation and development of pseudoarthrosis.	*S. aureus* 2.3 × 10^6^ CFU/mL 4.3 × 10^2^ CFU/mL	MRSA Amoxicillin/clavulanate CiprofloxacinColistin Trimethoprim/sulfamethoxazole	Five sessions of O_2_-O_3_-MAHT and ozone microinjections (3,2)	25 June 2019. Reduction in the oedema, joint effusion and pseudo-arthrosis symptoms. Reduction in inflammatory biomarkers.
4	RB	4 November 1963	13 June 2022. Mycobacterium infection, with BAL fluid positive for alveolar macrophages and other innate cells. Grocott Ziehl–Neelsen test for cytomegalovirus was negative. The patient suffered from a cough and hot chest	*S. aureus* 6.9 × 10^5^ CFU/mL 1.1 × 10^2^ CFU/mL	MRSA Amikacin Linezolid Moxifloxacin	Five sessions of O_2_-O_3_-MAHT (1)	5 September 2022. Reduction in inflammatory foci.
5	ZB	16 September 1954	18 February 2022. Post-surgery retro-peritoneum, para-aortic effusion with Gram-positive cocci due to septic infection of aortic prosthesis; CRP = 111 mg/dl.	*S. aureus* 7.3 × 10^6^ CFU/mL 2.0 × 10^3^ CFU/mL	MRSA Penicillin G Amoxicillin/clavulanate	Five sessions of O_2_-O_3_-MAHT (1,4)	19 May 2022. Reduction in sepsis via PET (SUVmax = 6.93 vs. 8.48).
6	AC	18 October 1972	24 August 2020. Surgery on hip arthroprosthesis. MRSA infection; CRP = 98 mg/dL, α1-globulin = 3.2, β1-globulin = 10.2	*S. aureus* 4.8 × 10^8^ CFU/mL 3.0 × 10^3^ CFU/mL	MRSA	Five sessions of O_2_-O_3_-MAHT (1)	11 November 2020. Strong reduction in inflammation.

(1) 50 μg/mL O_3_ in 200 mL of autologous blood; (2) (1) + 5 μg O_3_ in situ microinjections; (3) 40 μg/mL O_2_-O_3_-MAHT + 10 μg/mL O_3_ urethral insufflation; (4) 2 0_2_-0_3_-MAHT/week for 5 weeks with microinjections of 5 μg O_3_ in 20 mL + rectal insufflation of 20 μg O_3_ into 100 mL and ozone cream.

## Data Availability

Data can be easily requested from the corresponding author.

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
