# Peer review of "The Oxygen–Ozone Adjunct Medical Treatment According to the Protocols from the Italian Scientific Society of Oxygen–Ozone Therapy: How Ozone Applications in the Blood Can Influence Clinical Therapy Success via the Modulation of Cell Biology and Immunity"

_biology, 2023, doi:10.3390/biology12121512_

Round 1
Reviewer 1 Report
Comments and Suggestions for Authors There is no mention of the type of study (it is mandatory at least in the title). The nature of the study should be identifiable. The manuscript's content looks like an overview of the efficacy of ozone in the treatment of various medical conditions. Please refocus the objective of the study, otherwise, the entire piece would lack scientific significanceAuthor Response
There is no mention of the type of study (it is mandatory at least in the title). The nature of the study should be identifiable. The manuscript's content looks like an overview of the efficacy of ozone in the treatment of various medical conditions. Please refocus the objective of the study, otherwise, the entire piece would lack scientific significance
AUTHORS' RESPONSE:
Authors’ response: Right. The types of study and the objectives are reported, accordingly
Reviewer 2 Report
Comments and Suggestions for Authors
The manuscript entitles “The oxygen-ozone adjunct medical treatment according to the SIOOT protocols. How ozone in the blood is able to modulate cell biology and
immunity for clinics” documented a series of evidenced supported the use of ozone ad adjuvant in medicine.
However, different criteria should be considered after its final publication.
Main: The manuscript is a mixture of a review / research paper. The clinical trial described form the point 5 in ahead, should be presented in a separated manuscript.
The title does not reflect exactly the content of the manuscript and should be re-write. The use of acronyms in the title should be avoided.
Minor: The use of acronyms thought the text should be extensively revised. E.g. HNE line 265, 317.., OA, line 284, 286, 293..; MAHT line 47, 184, 241…; SIOOT, line 447, 530…
Paragraph in line 491-496 should be re-written, the interaction of O3 with Bood does not follow the Henry low.
Comments on the Quality of English LanguageMinor
Author Response
Reviewer #2
The manuscript entitles “The oxygen-ozone adjunct medical treatment according to the SIOOT protocols. How ozone in the blood is able to modulate cell biology and
immunity for clinics” documented a series of evidenced supported the use of ozone ad adjuvant in medicine.
However, different criteria should be considered after its final publication.
Main: The manuscript is a mixture of a review / research paper. The clinical trial described form the point 5 in ahead, should be presented in a separated manuscript.
Authors response: The paper was revised accordingly and transformed in a Research Article
The title does not reflect exactly the content of the manuscript and should be re-write. The use of acronyms in the title should be avoided.
Authors’ response: The title was revised accordingly
Minor: The use of acronyms thought the text should be extensively revised. E.g. HNE line 265, 317.., OA, line 284, 286, 293..; MAHT line 47, 184, 241…; SIOOT, line 447, 530…
Authors’ response: an acronyms paragraph was added in the text, accordingly
Paragraph in line 491-496 should be re-written, the interaction of O3 with Bood does not follow the Henry low.
Authors’ response: The ozone produced for ozone autohaemotherapy is in a gas form and at room temperature. Any gas introduced into a liquid (water, such as plasma) and depending on the medium temperature, should follow the Henry’s law, to evaluate precisely the content of ozone actually present in the plasma once ozone is injected in this medium from the gaseous state in the syringe. So, we do not understand this Reviewer’s comment.
Reviewer 3 Report
Comments and Suggestions for Authors
The manuscript describes the biological effects of medical oxygen-ozone (O2/O3) gas mixture on microbial and inflammatory induced disease pattern. It reviews the outcome and underlying mechanism of an oxygen-ozone major autohaemotherapy (O2-O3-MAHT) following the Scientific Society of Oxygen Ozone Therapy (SIOOT). The manuscript nicely combines overviews of the recent literature for different diseases with own research data. The manuscript pleasantly combines reviews of the current literature on the therapeutic effects of adjuvant ozone therapy with own research data regarding O2-O3-MAHT.
Enclosed you will find a few comments that should be considered.
Titel of the manuscript; The second part (“How ozone in the blood is able to modulate cell biology and immunity for clinics.”) makes for me no sense. What do the authors mean by cell biology and immunity in clinics? The following formulation would be more precise: (“How ozone applications in the blood can influence clinical therapy success via the influence on cell biology and immunity.”)
Fig 3 and legend: In the figure legend, the schema is divided into 3 parts (a, b, c). These letters do not appear in the schema itself. Here the letters should appear accordingly or be omitted in the legend.
Figure 4: The p-values are given here. As in Fig. 2, it is more harmonious to replace the values with **** asterisks. The actual p-values appear again in the text.
Figure 4: The font sizes in the individual subgraphs should be the same. The numbering with 1, 2, 3 at the right edge of the figure is unusual. I would recommend labeling the individual graphs from A to E and marking the individual box-plot pairs with 1 to 4 in graph E (Laquesne functional index).
Line 523: (50 μg/ml in 200 ml autologous blood). Here the authors give a concentration of the ozone but how many ml were given in the 200ml blood, what was the total amount of O3 in the 200 ml blood?
Author Response
The manuscript describes the biological effects of medical oxygen-ozone (O2/O3) gas mixture on microbial and inflammatory induced disease pattern. It reviews the outcome and underlying mechanism of an oxygen-ozone major autohaemotherapy (O2-O3-MAHT) following the Scientific Society of Oxygen Ozone Therapy (SIOOT). The manuscript nicely combines overviews of the recent literature for different diseases with own research data. The manuscript pleasantly combines reviews of the current literature on the therapeutic effects of adjuvant ozone therapy with own research data regarding O2-O3-MAHT.
Enclosed you will find a few comments that should be considered.
Titel of the manuscript; The second part (“How ozone in the blood is able to modulate cell biology and immunity for clinics.”) makes for me no sense. What do the authors mean by cell biology and immunity in clinics? The following formulation would be more precise: (“How ozone applications in the blood can influence clinical therapy success via the influence on cell biology and immunity.”)
Authors’ response: Done, accordingly
Fig 3 and legend: In the figure legend, the schema is divided into 3 parts (a, b, c). These letters do not appear in the schema itself. Here the letters should appear accordingly or be omitted in the legend.
Authors’ response: Figure 3 was revised accordingly
Figure 4: The p-values are given here. As in Fig. 2, it is more harmonious to replace the values with **** asterisks. The actual p-values appear again in the text.
Authors’ response: Figure 4 p values were replaced with asterisks, accordingly
Figure 4: The font sizes in the individual subgraphs should be the same. The numbering with 1, 2, 3 at the right edge of the figure is unusual. I would recommend labeling the individual graphs from A to E and marking the individual box-plot pairs with 1 to 4 in graph E (Laquesne functional index).
Authors’ response: Figure 4 revised accordingly. In the revised text Figure 4 is Figure 3 and Figure 3 is Figure 4
Line 523: (50 μg/ml in 200 ml autologous blood). Here the authors give a concentration of the ozone but how many ml were given in the 200ml blood, what was the total amount of O3 in the 200 ml blood?
Authors’ response: A sentence about this was reported in the text, accordingly